# Enhanced Water Age Performance Assessment in Distribution Networks

**Laura Monteiro** [1] , **Ricardo Algarvio** [2] **and Dídia Covas** [1,*]

1    CERIS, Instituto Superior Técnico, Universidade de Lisboa, 1049-001 Lisbon, Portugal;
laura.monteiro@tecnico.ulisboa.pt
2    SENER, Ingeniería y Sistemas SA, Cerdanyola del Vallès, 08290 Barcelona, Spain; ricardo.algarvio@gmail.com
*    Correspondence: didia.covas@tecnico.ulisboa.pt; Tel.: +351-218-418-152

**Abstract:** Water age is frequently used as a surrogate for water quality in distribution networks and is often included in modelling and optimisation studies, though there are no reference values or standard performance functions for assessing the network behaviour regarding water age. This paper presents a novel methodology for obtaining enhanced system-specific water age performance assessment functions, tailored for each distribution network. The methodology is based on the establishment of relationships between the chlorine concentration at the sampling nodes and simulated water age. The proposed methodology is demonstrated through application to two water distribution systems in winter and summer seasons. Obtained results show a major improvement in comparison with those obtained by published performance functions, since the water age limits of the performance functions used herein are tailored to the analysed networks. This demonstrates that the development of network-specific water age performance functions is a powerful tool for more robustly and reliably defining water age goals and evaluating the system behaviour under different operating conditions.

**Keywords:** water age; water quality; water distribution systems; performance functions; free chlorine

## 1. Introduction

Water age is defined as the time taken for the water to travel from the source to the consumption locations within the distribution system [1]. Water age depends on the pipe lengths and diameters, as well as on the water consumption at the nodes, varying along the water distribution network (WDN). The water flow paths, determined by the distribution system layout, valve settings and pump operation, also affect the water traveling time. Water age can vary from a few hours to several days, due to daily and seasonal variations in water demand [2]. Water age at a given location of a WDN cannot be directly measured and has to be inferred from a tracer test or computed using hydraulic software for WDN simulation [3].

While water travels through the WDN, many chemical and microbiological reactions occur, changing the quality of the water at the consumers' tap and potentially compromising public health [4,5]. Chlorine residual concentrations decline with increasing travel time (water age), which is usually followed by an increase in bacterial counts and diversity [6,7]. The extent of such reactions increases with water age; for this reason, this parameter is considered a useful indicator of the quality of the water [8,9] and has been used as a surrogate for water quality in many WDN studies [10,11].

However, water quality degradation in the networks also depends on other factors, such as the pipe material and physical degradation, the water temperature [12] and the upstream treatment [2]. Hence, and despite correlations between mean water age and water quality in a WDN having been observed [8,9], water age is of little value as a surrogate for a specific microbial water quality parameter.

Many efforts have been made to reduce water age in order to minimise water quality degradation in drinking water networks, whether by rerouting flow paths, through valve operation [13], by increasing nodal outflows at critical dead-end nodes (through the opening of a blow-off at the hydrant site) [14], or by increasing storage tanks' daily turnover rates to reduce water ageing in storage facilities [15]. In water quality optimisation studies, the objective is often to minimise water age at the consumption nodes without setting water age goals [10,11]. Consequently, the benefit of the optimisation can only be assessed as a percentage of water age decrease. The extent of that decrease regarding water quality improvement or compliance with water quality standards is still unknown, since no reference values for water age in distribution systems exist.

The first water age performance function (also known as a penalty curve) was developed by Coelho (1996) [16]. This author proposed a new performance index, PI, ranging from 0 (no service) to 1 (optimum performance), defined as follows:

$$
\begin{aligned}
&PI = 1; \text{ if } WA \leq 6 \text{ h} \\
&PI = -0.125 \times (WA - 6) + 1; \text{ if } 6 \text{ h} < WA < 10 \text{ h} \\
&PI = 0; \text{ if } WA \geq 10 \text{ h}
\end{aligned}
\tag{1}
$$

in which WA is the water age at the nodes (in hours). Accordingly, if the water age is less than or equal to 6 h, the performance of the system is maximum. The performance index decreases linearly with time towards 0.5, when water age increases from 6 to 10 h, above which the performance index is null. The water age limits were set based on a single case study analysed in Edinburgh, the U.K.

Later, Tamminen et al. (2008) [17] proposed three different water age performance curves, differing from the previous one in the optimal and unacceptable performance limits, ranging from 10 to 50 h (lower limits) and from 30 to 350 h (upper limits). The authors provided no explanation for the proposed limits. The developed functions were applied to the benchmarking network Net3 for two operating scenarios and clarified that the performance regarding water age is mostly dependent on the usage of the tanks, where water age increases the most.

Shokoohi et al. (2017) [18] proposed the performance function described by:

$$
\begin{aligned}
&PI = 1; \text{ if } WA \leq 8 \text{ h} \\
&PI = -0.025 \times (WA - 8) + 1; \text{ if } 8 \text{ h} < WA < 48 \text{ h} \\
&PI = 0; \text{ if } WA \geq 48 \text{ h}
\end{aligned}
\tag{2}
$$

The water age limits of 8 and 48 h were specified based on studies on the effect of water residence time on total bacteria in the bulk water, carried out in a laboratory pipe rig [19]. This performance function sets higher water age upper limits for each interval than does that proposed by Coelho (1996). This performance function was combined with a free chlorine performance function for the development of a water quality reliability index.

Recently, a new water age performance function was proposed for service reservoirs within a water distribution network [20], as in Equation (3):

$$
\begin{aligned}
&PI = -0.0188 \times WA + 1; \text{ if } WA < 48 \text{ h} \\
&PI = 0.1; \text{ if } WA \geq 48 \text{ h}
\end{aligned}
\tag{3}
$$

Unlike the previous functions, this performance function decreases linearly with time since the water enters into the system until it reaches a close-to-null value at 48 h. The authors combined this function with analogous chlorine and trihalomethanes (THM) performance functions as subindices of a water quality index. The index was used to compare the performance of service reservoirs regarding water quality in optimisation studies. Both Shokoohi et al. (2017) and Nyirenda and Tanyimboh (2020) [20] agree that the performance is minimum when the water age is above 48 h.

The water age performance curves proposed in the literature are presented in Figure 1. The main differences between the authors are the upper and the lower limits of maximum

and minimum performance. Coelho's (1996) curve has a significantly lower upper limit (10 h instead of 48 h). These curves result from the application experience of each author, demonstrating that these limits can be very case-specific and require case-fitting. Due to the dependency of water quality degradation reactions on local factors (e.g., content and nature of dissolved organic matter, pipe conditions, water temperature), the above-described performance functions for water age are very unlikely to be appropriate for universal application. However, these specific functions can be very useful for assessing optimisation solutions and the overall water quality performance of a WDN. A key question remains as to how water age performance functions can be determined for each distribution network in a systematic and robust way.

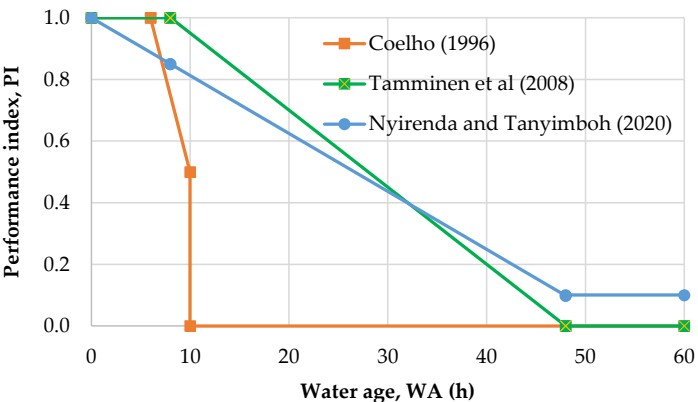

**Figure 1.** Comparison of the existing water age performance curves.

In this paper, a novel methodology for developing water age performance assessment curves tailored for each water distribution network is proposed and demonstrated with two case studies. The proposed methodology, based on the analysis of available water quality data and on hydraulic modelling, allows the development of system-specific performance functions and the setting of water age goals that can be used to include water quality in optimisation studies and in the performance assessment of a WDN.

An innovative systematic approach to obtain enhanced water age performance assessment functions specific to each WDN is presented by combining a free chlorine concentration performance function (defined based on acceptable concentration limits) with the relationship between water age and free chlorine concentration obtained based on measurements carried out in the WDN. This is the first time that such a physically based approach (based on chlorine concentration limits) has been proposed to obtained water age performance curves.

## 2. Methodology

### 2.1. General Approach

The novel methodology for developing case-specific water age performance functions is based on field data collection, processing and analysis, as well as on the numerical simulation of the WDN behaviour. The approach comprises five main steps:

(1) Physical and operational data collection and model development.
(2) Water quality data collection, processing and analysis.
(3) Water age estimation based on numerical simulations.
(4) Development of correlations between water age and free chlorine concentration.
(5) Development of water age performance functions.

A detailed description of these main steps is presented in the following sections.

### 2.2. Physical and Operational Data Collection and Model Development (Step 1)

The data necessary for developing and calibrating the hydraulic models of the WDN, such as infrastructure characteristics, flow rate data, pressure monitoring data and billing

data, are primarily collected. A network hydraulic model is then developed by using, for instance, EPANET [1], including both the physical and the operational conditions of the system (i.e., consumption, pump operating rules, and valve settings). Typically, WDN have two extreme operating periods, representative of winter and summer seasons. In Portugal, these correspond to the months of February (with the minimum daily consumption) and August (with the maximum daily consumption).

### 2.3. Water Quality Data Collection, Processing and Analysis (Step 2)

Water quality data in the two periods (winter and summer) over at least 5 years should be collected. Data should include the results of the microbiological and chemical analysis carried out by the utilities for the mandatory water quality control programme.

Data processing and analysis starts by identifying the nodes in the hydraulic model that correspond to the water sampling locations. The water quality datasets are further analysed in order to identify for which non-conservative parameters there are enough data for a correlation analysis with water age. Often, free chlorine is the parameter for which more data are available, being frequently monitored in most WDN. In chlorinated water systems, total chlorine may be the parameter that best represents the residual disinfectant content in the water. The water quality datasets may also contain results of non-conservative water quality parameters, such as trihalomethanes (THM), which are only seldom analysed. In addition, the results of microbiological analyses, such as for *E. coli*, coliform bacteria, and heterotrophic plate counts (HPCs), are mostly null, which hinders any attempt to find correlations between water age and those parameters. For those reasons, only free chlorine was selected for further correlation analysis with water age.

Chlorine records under the minimum detectable limit should be removed from the datasets. In Portugal, the DPD method is typically used; the corresponding minimum detectable limit is 0.05 mg/L. For each sampled location, the mean chlorine content is calculated in each season, unless there is high variability among the samples (>0.2 mg/L). In such a case, the older records should be discarded, as the chlorine concentration difference may be due to dissimilar operating conditions of the networks that are not representative of the simulated demand seasons.

### 2.4. Water Age Estimation (Step 3)

Water age at the sampling nodes is determined by running extended-period simulations of the WDN behaviour, using small quality time steps (1 min), for each characteristic season. Obtained results for the first hours of simulation, which show a linear increase in water age with time, should be discarded, as these correspond to the stabilisation of the water age in the model. The duration of this stabilisation period depends on the water demand and on the node location; thus, it must be determined for each case. The water age time series at each sampling node should then be examined for their variation over time in order to assess the suitability of using the mean value as the characteristic water age at the node, particularly by the time the grab samples are taken (usually from 9 a.m. to 5 p.m.). For those nodes where high variations in water age values are observed throughout the day, a clear correlation between water age and any water quality parameter might be unfeasible.

### 2.5. Development of Correlations between Water Age and Free Chlorine Concentration (Step 4)

Correlations between water age and free chlorine concentration at the nodes are assessed by plotting the two variables and evaluating their trend. Linear, exponential and polynomial trends can be observed and the respective regression functions must be obtained. The quality of the correlation is typically assessed by means of the coefficient of determination ($R^2$). Equations to predict the chlorine concentration as a function of water age are determined for each season.

### 2.6. Development of Water Age Performance Functions (Step 5)

Water age performance functions are developed based on the obtained correlations between the free chlorine concentration and water age, Cl = f(WA), combined with a chlorine performance function, PI = f(Cl). For a given water age, the expected chlorine concentration is estimated using the previously obtained function. The chlorine concentration is then converted into a performance index by making use of a general chlorine performance function. Distinct performance functions are developed for the winter and summer seasons.

Despite performance functions for the free chlorine concentration having been published in the literature [16], a general chlorine performance function is also proposed herein (Equation (4)), based on the current guidelines [21] and on the Portuguese law on drinking water quality [22]:

$$
\begin{aligned}
&PI = 5\,Cl;\ \text{for } Cl < 0.2\ \text{mg/L}\\
&PI = 1;\ \text{for } 0.2\ \text{mg/L} \le Cl \le 0.6\ \text{mg/L}\\
&PI = -0.7143\,Cl;\ \text{for } 0.6\ \text{mg/L} < Cl < 2.0\ \text{mg/L}\\
&PI = 0;\ \text{for } 2\ \text{mg/L} \le Cl \le 5\ \text{mg/L}
\end{aligned}
\tag{4}
$$

where Cl is the free chlorine concentration (mg/L) and PI is the performance index, ranging from 0 (no-service situation) to 1 (optimum performance). The proposed free chlorine performance function is depicted in Figure 2.

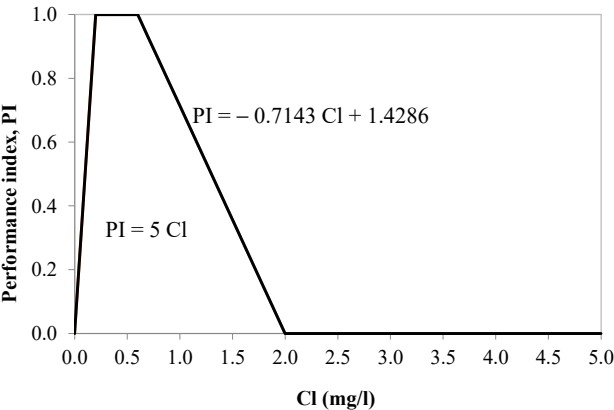

**Figure 2.** Proposed free chlorine performance function for drinking water distribution systems described by Equation (4).

The rationale of this penalty curve is the following. Optimum performance is assigned to chlorine concentrations within the 0.2 to 0.6 mg/L range. Under 0.2 mg/L, the minimum required concentration for ensuring protection of the drinking water [21], the performance decreases linearly with decreasing chlorine level. Above 0.6 mg/L, the chlorine concentration promotes increased DBP formation and complaints due to taste and odour. Thus, the performance linearly decreases with increasing chlorine concentration until 2.0 mg/L, which is the chlorine threshold for odour in distilled water [23]. Performance is null in the range of 2.0 to 5.0 mg/L, the latter being the guideline value for free chlorine in drinking water [21]. According to the developed function, the performance associated to a given node of a WDN where the chlorine concentration is above 0.6 mg/L and up to 1 mg/L (which is quite common, especially near the treatment plants) is high ($\ge 0.7$) but not optimum, as the upper limit of the recommended chlorine concentration range is exceeded.

## 3. Case Studies

### 3.1. Case Study 1

Case Study 1, the Costeira water distribution network, supplies part of the city of Castelo Branco, in Portugal (Figure 3). This network comprises one storage tank and 116 km of pipes that serve approximately 5900 branches. The network pipe materials are polyvinyl chloride (PVC) (93%) and ductile iron. The network is divided into 12 district-metered areas (DMAs). The supplied water volume in 2019 was 1.1 Mm$^3$ to domestic and industrial water users. The water demand in August 2020 (152,906 m$^3$) was approximately 50% higher than that in February (96,969 m$^3$), mostly due to the floating population and the irrigation of urban gardens. Two main seasons are clearly identifiable in this WDN. The network has two large consumers: a dairy products factory and a hospital. The factory has an associated DMA with no other consumers and is located in the southern industrial area. The hospital is connected to a water main, very close to the water tank that supplies the Costeira subsystem, so its influence on the water age in the rest of the network is negligible.

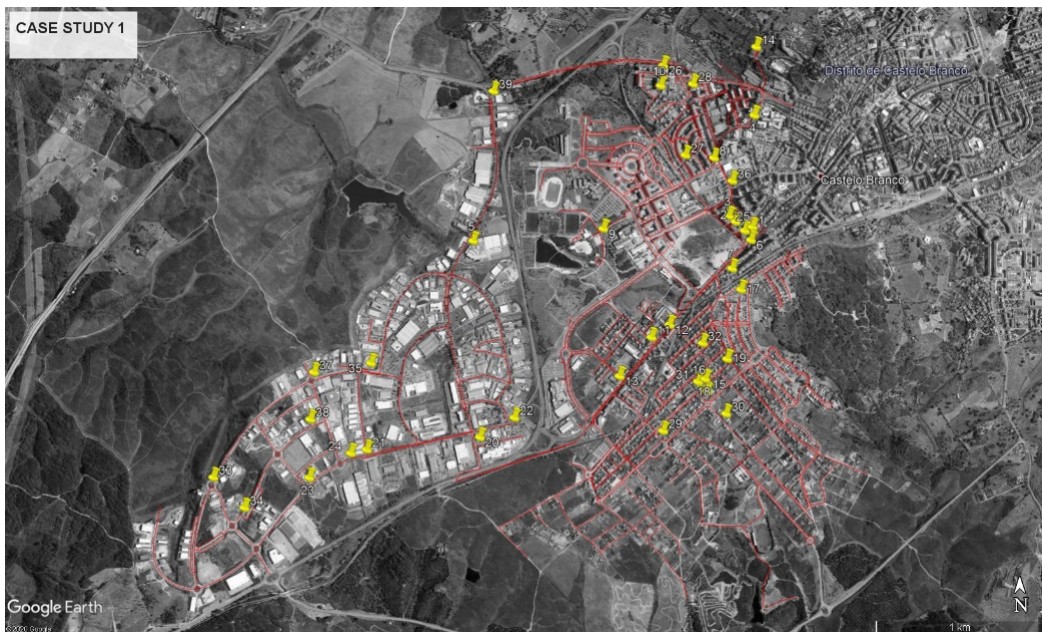

**Figure 3.** Case Study 1 water distribution system and water quality sampling locations.

### 3.2. Case Study 2

Case Study 2, the water distribution network of Quinta do Lago, located in south Portugal, supplies 1.7 Mm$^3$/year of water mainly to domestic consumers and hotels (Figure 4). The system is located in a touristic area, and the water demand in summer is 4.5 times the demand in winter. The network comprises one single storage tank and 72.8 km of pipes, ranging from 63 to 400 mm in diameter. The predominant pipe materials are PVC (53%) and asbestos cement (44%). Water consumption is measured by a telemetry system every hour at each of the 2000 clients. Pressure is measured every minute at seven locations within the distribution system.

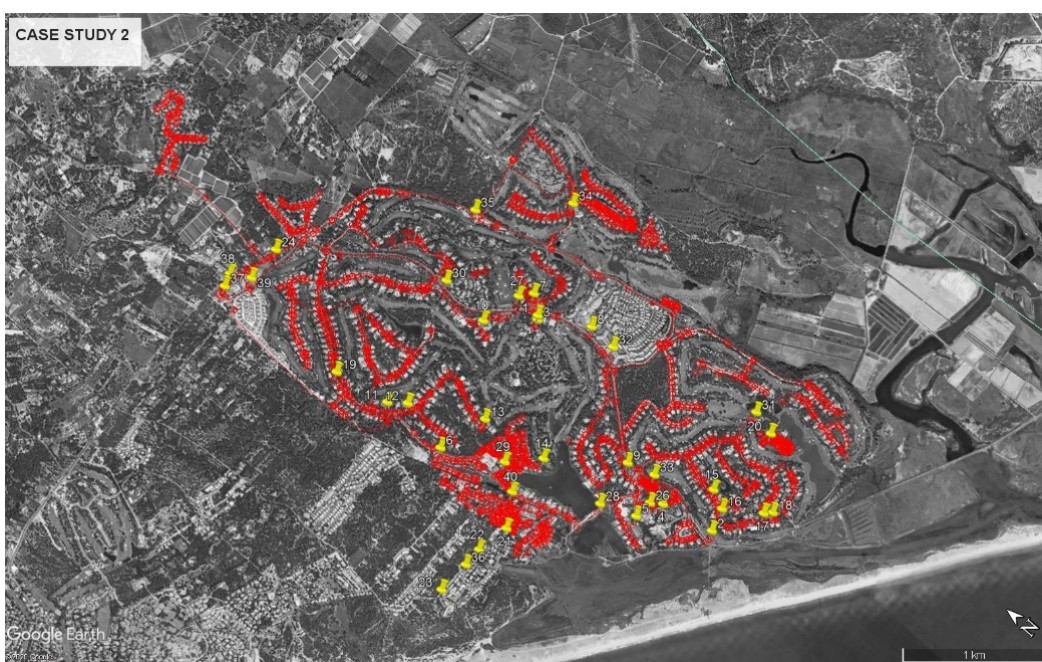

**Figure 4.** Case Study 2 water distribution system and water quality sampling locations.

## 4. Methodology Demonstration

The proposed methodology was applied to two water distribution systems for two seasons (winter and summer). The developed functions were further applied to evaluate the water age in the two systems.

### 4.1. Physical and Operational Data Collection and Model Development

The first steps of the proposed methodology include data collection and analysis. Two types of data were collected, namely, physical and operational data of the WDNs, necessary to develop the hydraulic simulator, as described in Section 3. The respective mathematical models were developed using EPANET hydraulic simulator. Because the methodology requires a good estimation of the water age at the nodes, a sound calibration of the models is necessary. For Case Study 1, the model results for flow rate and pressure at the entrance nodes of each DMA were compared with the historical records. A correction factor was then applied to each nodal demand according to its DMA, in order to match the observed and simulated pressure and flow at the nodes. For Case Study 2, flow rate and pressure online data acquired at locations within the network were used for the model validation. Model calibration was carried out for each season.

### 4.2. Water Quality Data Collection, Processing, and Analysis

Water quality samples over a 3–5-year period were collected, processed, and analysed over several-year periods in the two case studies. These data were collected from the water utilities' records and are the results of periodical sampling and analysis of grab samples, collected from consumers' taps at different locations in the network.

In Case Study 1, the collected water quality data during a 5-year period (2015–2019) show that the results for the microbiological parameters were mostly null and that free chlorine was the parameter that was most frequently measured. In the winter period (February), a total of 34 locations within the Costeira system were sampled for chlorine concentration measurement over 5 years. A total of 278 samples were analysed, of which 168 were taken at the hospital node and the remaining 110 samples were collected at 33 nodes across the network. In summer (August), only 11 locations were sampled and a total of 138 samples were analysed. Of those, 71 samples were collected at the hospital and 67 samples were collected at other nodes across the network. In winter, the chlorine concentration in 32% of the samples (excluding the hospital samples) was within the

optimum range (0.2 to 0.6 mg/L, according to national statutes), 60% were above 0.6 mg/L, and only 9% were below the minimum recommended concentration (Figure 5a). In summer, about half of the samples had a chlorine concentration below the 0.2 mg/L limit, and almost no samples were in the optimum range, showing a large decrease in chlorine content over the seasons, which is most likely due to increased water temperatures.

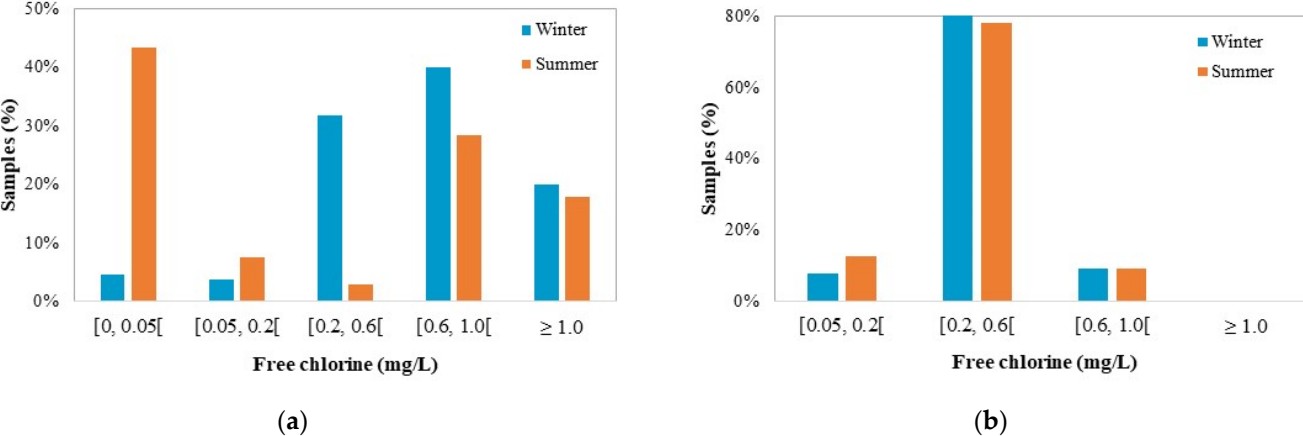

(**a**)   (**b**)

**Figure 5.** Chlorine concentrations at sampling nodes for (**a**) Case Study 1 and (**b**) Case Study 2.

In Case Study 2, the water quality dataset also showed that the microbiological parameters were null almost all the time, while free chlorine was the parameter that was most frequently measured. In the winter period, a total of 33 locations within the system were sampled for chlorine concentration measurement over 8 years (2008–2010, 2012–2015, 2017). A total of 65 samples were analysed for free chlorine. In the summer period, 41 locations were sampled and 87 samples were collected. The results of chlorine monitoring in the system show that about 80% of the samples had a chlorine content within the optimum range (0.2 to 0.6 mg/L) in both seasons (Figure 5b).

*4.3. Water Age Estimation*

Water age was estimated based on extended-period simulations of the WDNs using the developed EPANET models. In both case studies, two extended-period simulations were carried out to determine the water age at the sampling nodes: one for winter and another for summer operating conditions.

In Case Study 1, the duration of the extended-period simulations was 696 h for both seasons. For winter, the water age at approximately 75% of the nodes was less than 24 h, and 86% of the nodes received water that entered the system in the last 48 h. In the summer, those percentages increased to 87% and 91%, respectively, for 24 and 48 h.

In Case Study 2, simulations were carried out for 11 days (264 h). In winter, the water age was less than or equal to 24 h in 18% of the nodes, while it was less than or equal to 48 h in 58% of the nodes. In summer, these percentages increased to 90% and 96%, respectively, for 24 h and 48 h, as a consequence of the increase in water demand.

*4.4. Development of Correlation between Water Age and Free Chlorine Concentration*

The chlorine variation with water age was determined for both case studies and for each season, based on collected chlorine data (Step 2) and on the water age estimated by extended-period simulations (Step 3) at the same nodes. The results show that the mean chlorine concentration at the sampled nodes generally decreased with increasing water age for both case studies and in both seasons (Figures 6 and 7). Linear regressions were fitted to obtain relationships to describe the variation in water age with the chlorine concentration.

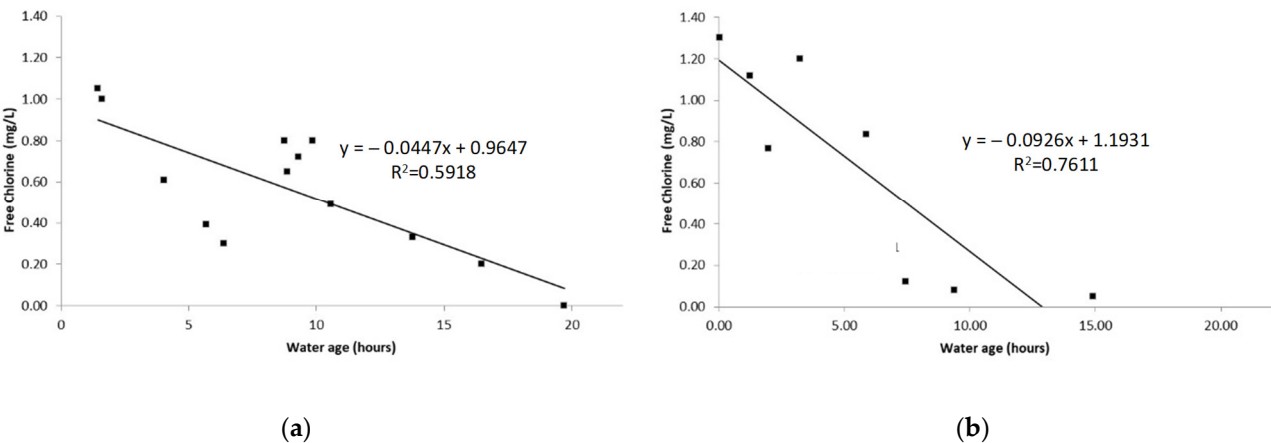

**Figure 6.** Mean water age and chlorine concentrations at the sampling nodes in Case Study 1 in (**a**) winter and (**b**) summer.

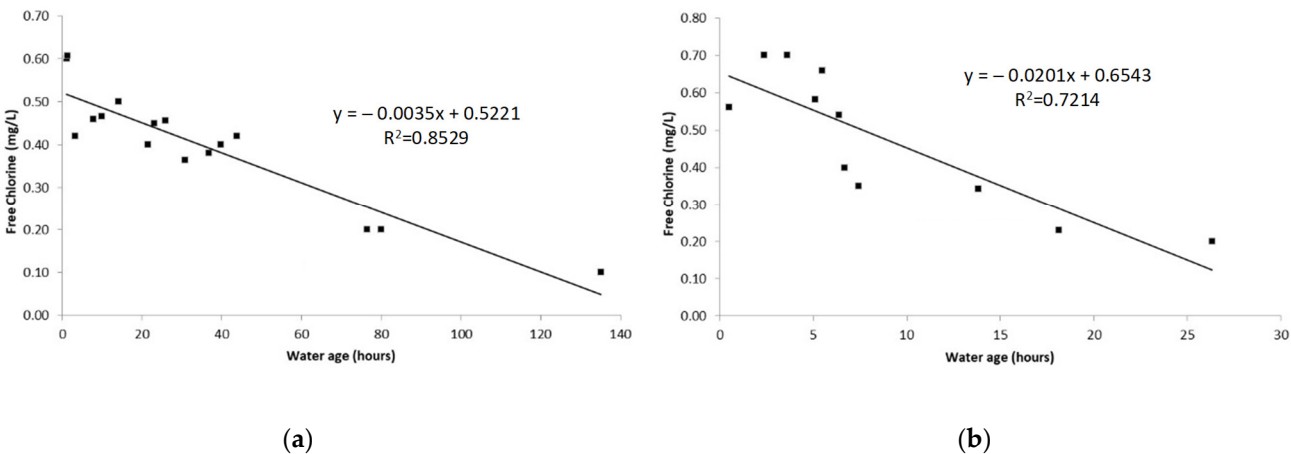

**Figure 7.** Mean water age and chlorine concentrations at the sampling nodes in Case Study 2 in (**a**) winter and (**b**) summer.

For Case Study 1, the coefficients of determination of the linear regressions, $R^2$, were 0.59 and 0.76 in winter and in summer, respectively, which demonstrates a close-to-linear relationship between the mean water age and free chlorine concentration at the nodes. The comparison between the two equations obtained for the winter and summer seasons (Figure 6) shows that the slope was slightly higher in summer than in winter, which is in agreement with faster chlorine decay in summer. Consequently, the chlorine concentration reaches close-to-zero values in about 13 h in summer, while it takes 22 h in winter. Thus, the water age can be related to the chlorine concentration in the Case Study 1 system by the following Equations (5) and (6):

$$Cl = -0.0447 \, WA + 0.9647 \text{ for WA} < 22 \text{ h; winter} \qquad (5)$$

$$Cl = -0.0926 \, WA + 1.1931 \text{ for WA} < 13 \text{ h; summer} \qquad (6)$$

where Cl is the free chlorine concentration (mg/L) and WA is the mean water age (h) at the nodes during the day. For higher WA values, the expected chlorine concentration is null.

For Case Study 2, good correlations between the mean water age and mean chlorine concentrations at the nodes were also obtained, since the coefficients of determination of the linear regressions were 0.85 in winter and 0.72 in summer (Figure 7). The slope of the obtained equation was also higher in summer than in winter. Consequently, chlorine decays faster and reaches close-to-zero values in about 33 h in summer, while it takes 149 h in winter. Thus, the water age can be related to the chlorine concentration in the Case Study 2 system by the following Equations (7) and (8).

$$Cl = -0.0035 \text{ WA} + 0.5221 \text{ for WA} < 149 \text{ h; winter} \tag{7}$$

$$Cl = -0.0201 \text{ WA} + 0.6543 \text{ for WA} < 33 \text{ h; summer} \tag{8}$$

For WA higher than 149 h, the expected chlorine concentration is null.

### 4.5. Development of Water Age Performance Functions

Water age performance functions were developed by combining the previously developed correlations with the general free chlorine performance function for water distribution systems described by Equation (4) (Figure 2). The obtained penalty functions are described by Equations (9) and (10) for winter and summer in Case Study 1. These penalty curves correspond to those described by Equation (4) in which the chlorine concentration is replaced by the previously obtained correlations (Figure 6); as a result, the performance index, PI, becomes a function of WA instead of a function of the chlorine concentration.

$$\begin{aligned} &PI = 0.0319 \text{ WA} + 0.7395; \text{ WA } \le 8 \text{ h} \\ &PI = 1; 8 \text{ h} < \text{WA} \le 17 \text{ h} \\ &PI = -0.2067 \text{ WA} + 4.5078; 18 \text{ h} < \text{WA } \le 21 \text{ h} \\ &PI = 0; \text{ WA} > 21 \text{ h} \end{aligned} \tag{9}$$

$$\begin{aligned} &PI = 0.0353 \text{ WA} + 0.723; \text{ WA } \le 9 \text{ h} \\ &PI = 1; 9 \text{ h} < \text{WA} \le 14 \text{ h} \\ &PI = -0.393 \text{ WA} + 6.4355; 14 \text{ h} \le \text{WA } 16 \text{ h} \\ &PI = 0; \text{ WA} > 16 \text{ h} \end{aligned} \tag{10}$$

Similarly, for Case Study 2, water age performance functions were developed for winter Equation (11) and for summer Equation (12).

$$\begin{aligned} &PI = 1; \text{ WA } \le 93 \text{ h} \\ &PI = -0.0175 \text{ WA} + 2.6105; 93 \text{ h} < \text{WA } \le 147 \text{ h} \\ &PI = 0; \text{ WA} > 147 \text{ h} \end{aligned} \tag{11}$$

$$\begin{aligned} &PI = 0.0144 \text{ WA} + 0.9612; \text{ WA } \le 2 \text{ h} \\ &PI = 1; 2 \text{ h} < \text{WA} \le 22 \text{ h} \\ &PI = -0.1005 \text{ WA} + 3.2715 ; 22 \text{ h} < \text{WA } \le 32 \text{ h} \\ &PI = 0; \text{ WA} > 32 \text{ h} \end{aligned} \tag{12}$$

The four performance functions (Figure 8) differ from those in the literature in their shape and water age limits. The developed functions for Case Study 1 (and for Case Study 2 in summer) penalise very short residence times, as these correspond to chlorine concentrations higher than the maximum recommended concentration (0.6 mg/L). Thus, the water age performance is not optimum in the first hours, as in the functions by Coelho (1996) and Shokoohi et al. (2017).

For Case Study 2 in winter, the water age performance is optimum for all water ages less than 93 h, and the shape of the function is similar to those in literature, differing only in the upper limit. This is due to the lower chlorine concentrations in the water that is supplied to the Case Study 2 network.

The performance functions for Case Study 1 and for Case Study 2 in summer also have in common a water age limit that is higher than the 10 h proposed by Coelho (1996) and less than the 48 h of Shokoohi et al. (2017). The water age limits associated to null water age performance are 21 h and 16 h in Case Study 1 (winter and summer, respectively) and 32 h in Case Study 2 in summer. In addition, for Case Study 2 in winter, it takes about 147 h for the performance index to reach null values, demonstrating that the maximum recommendable water age in a distribution system must be determined on a case-by-case basis.

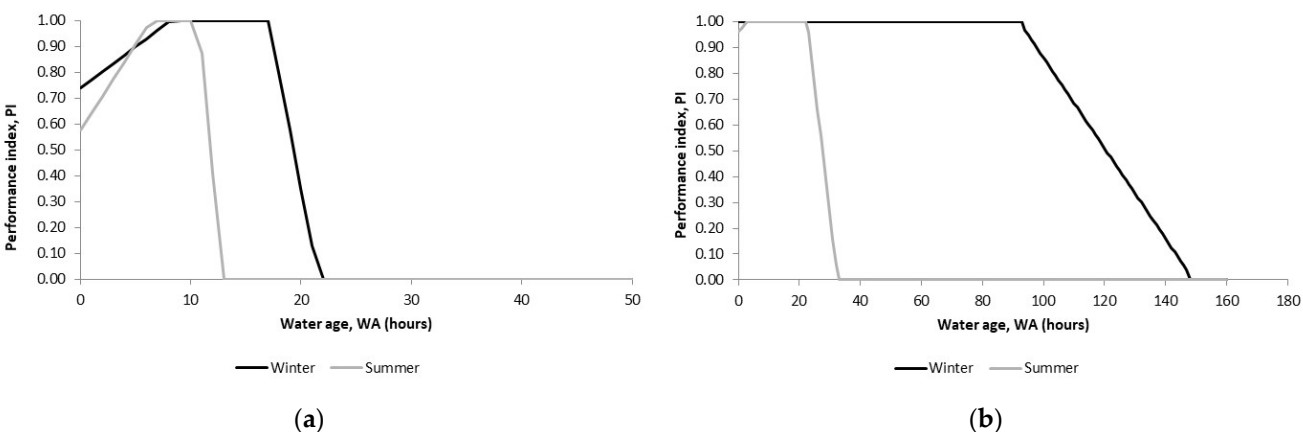

**Figure 8.** Water age performance curves for (**a**) Case Study 1 and (**b**) Case Study 2.

The differences among the obtained performance functions are mostly due to differences in chlorine stability in the water in each system. In Case Study 1, chlorine always decays faster than in Case Study 2, regardless of the season, which is likely due to differences in source water quality and treatment. That brings additional difficulty in managing chlorine levels in the Case Study 1 network (Figure 3). For Case Study 2, longer water ages do not seem to pose a problem for chlorine management in winter. This is likely due to the very low reactivity of the organics in the water that is supplied by a long upstream transmission system, where most of the chlorine decay reactions take place.

The water age limits in the performance functions allow us to set water age goals and to identify the nodes that are supplied with aged water. Nodes of higher water age are generally those located in peripheral areas and further away from the water source (see Figures 9 and 10). Identifying such nodes and the water age limits can be very useful for the optimisation of water distribution systems, by narrowing the optimisation problem to those nodes that effectively require water age minimisation.

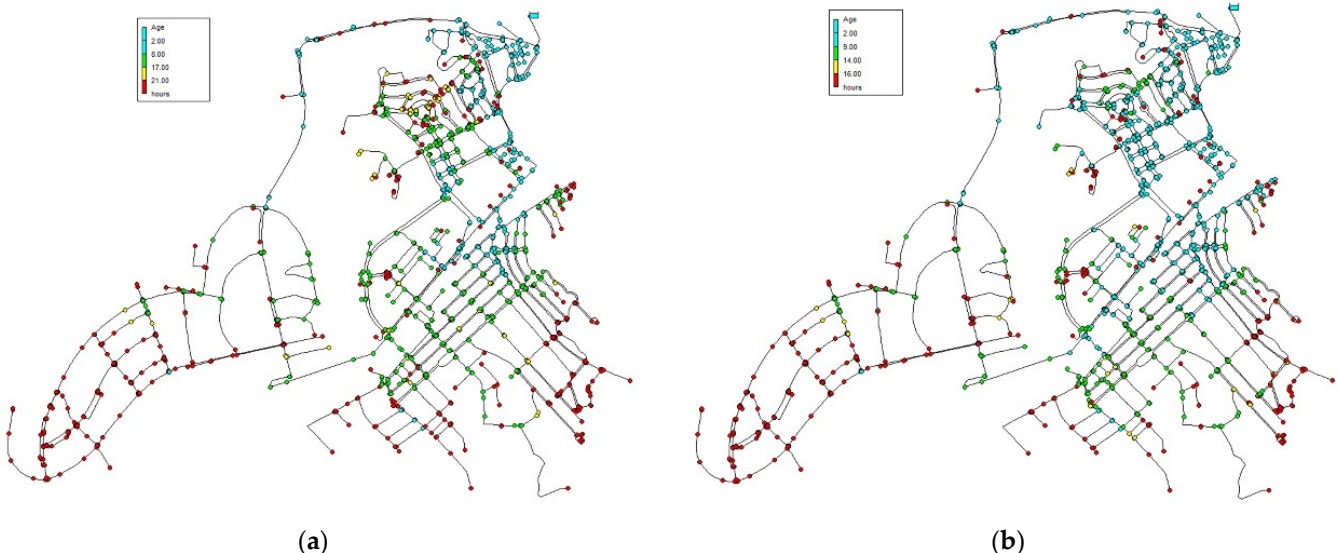

**Figure 9.** Water age distribution over the network in Case Study 1 (**a**) in winter and (**b**) in summer.

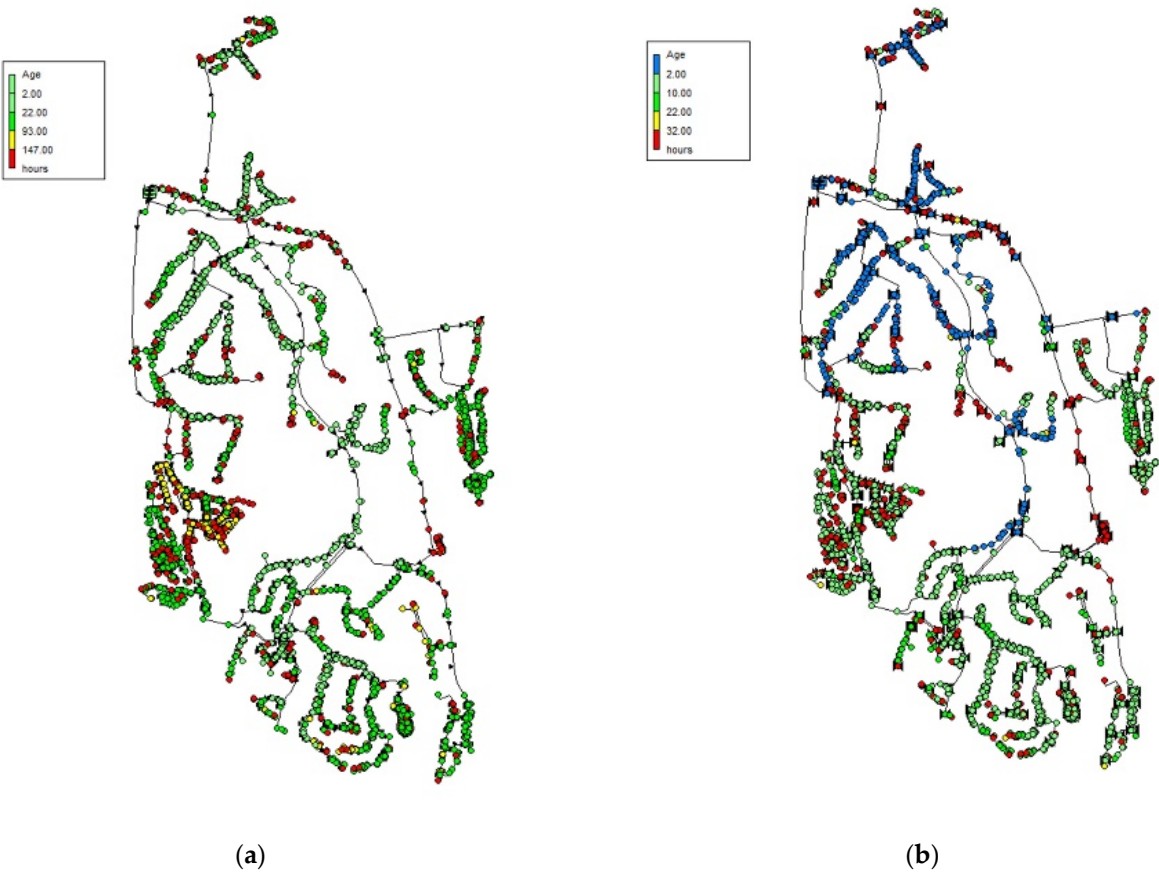

(**a**)                                                                    (**b**)

**Figure 10.** Water age distribution over the network in Case Study 2 (**a**) in winter and (**b**) in summer.

## 5. Results Application and Discussion

Water age performance in the two case studies was further assessed by applying the obtained functions to the consumption nodes in each WDS over a 24 h period. The water age at each consumption node at each hour was converted into a performance index (PI) from 0 to 1 by using Equations (9)–(11) or (12), and the results are summarised in Figure 11. Performance functions from the literature were also applied in the same way for comparison. A mean performance index was also determined for a global evaluation of the network (Table 1). A classification scale for the performance index is depicted in Table 1 for the sake of comparison of results.

The application of the specific performance function to the Case Study 1 network in winter conditions shows that the performance index is higher than 0.75 at about 57% to 73% of the consumption nodes, depending on the hour of the day (Figure 11a). However, for 21% to 33% of the nodes, the performance is null, as a result of the water age being higher than the upper limit (21 h). In summer, the percentage of nodes that are supplied with good-quality water (PI > 0.75) increases (73% to 85% of the nodes), and only about 14% to 21% of the nodes are supplied with water that has been in the network for more than 16 h, for which the performance is null (Figure 11b). Accordingly, the global performance of the system increases from 0.68 in winter to 0.77 in summer (Table 1). These results demonstrate that the lower water demand in winter has a worse effect on water quality, by increasing travel times and water stagnation, than the temperature increase in summer, since it is counterbalanced by increased water consumption.

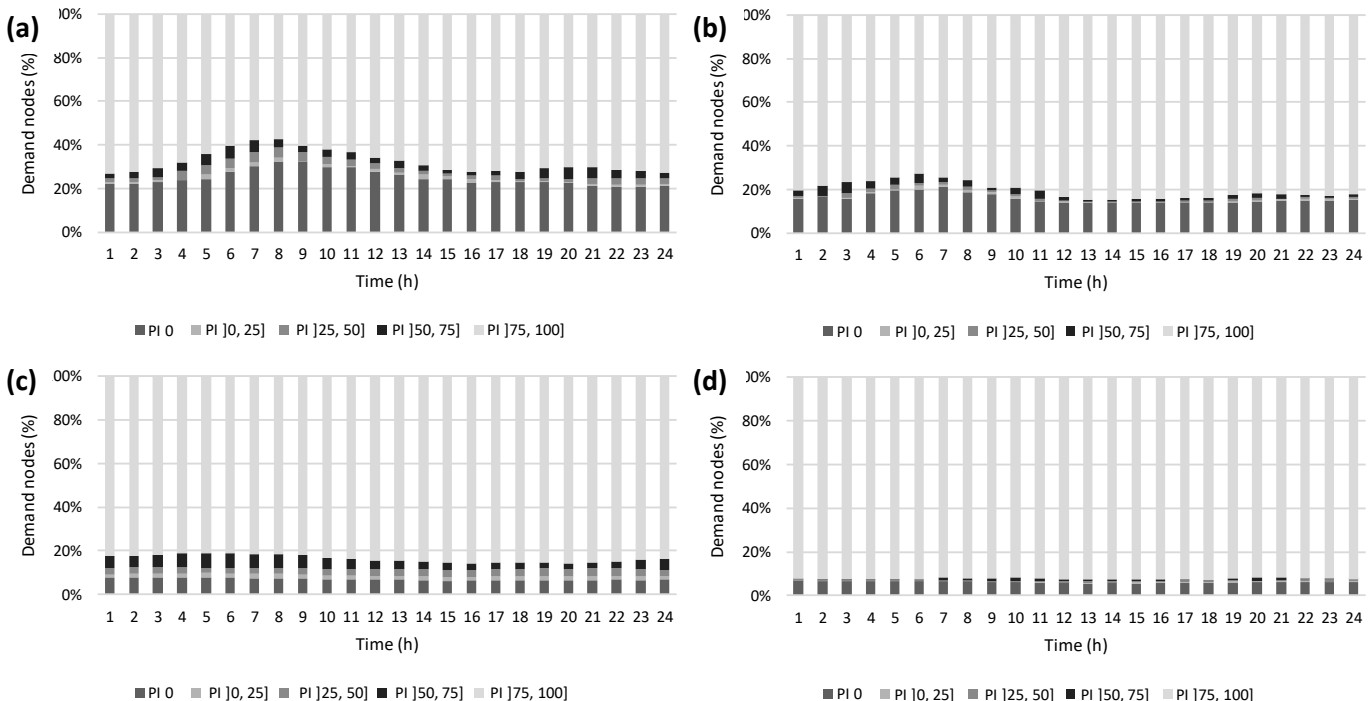

**Figure 11.** Performance index at the consumption nodes over a 24 h period in (**a**) Case Study 1 in winter, (**b**) Case Study 1 in summer, (**c**) Case Study 2 in winter, and (**d**) Case Study 2 in summer.

**Table 1.** Global performance index obtained using the proposed functions and those from the literature.

| | Mean Water Age (h) | Global Water Age Performance Index | | | |
|---|---|---|---|---|---|
| | | Proposed Function | Coelho (1996) [16] | Shokoohi et al. (2017) [18] | Nyirenda and Tanyimboh (2020) [20] |
| Case 1—winter | 41 | 0.64 ● | 0.32 ● | 0.72 ● | 0.62 ● |
| Case 1—summer | 22 | 0.74 ● | 0.55 ● | 0.84 ● | 0.74 ● |
| Case 2—winter | 53 | 0.93 ● | 0.02 ● | 0.27 ● | 0.22 ● |
| Case 2—summer | 14 | 0.91 ● | 0.46 ● | 0.86 ● | 0.74 ● |

Note: Assuming the following PI classification scale: ● good performance (>0.70); ● adequate performance (0.4 < PI ≤ 0.70); ● unacceptable performance (≤0.40).

For the Case Study 2 network, the performance regarding water age is better than that for Case Study 1, as the percentage of nodes with good performance is always higher than 81% and 92% in winter and in summer, respectively (Figure 11c,d). Only about 6–7% of the nodes are supplied with water older than the upper limit of the performance function. Consequently, the global performance index is higher for Case Study 2 than for Case Study 1 (Table 1).

These results obtained when applying system-specific water age performance functions are not in agreement with those obtained when applying the functions from the literature (Table 1), mainly because the water age limits in those functions are not adequate for the specifics of the analysed systems. In general, the Shokoohi et al. (2017) function generates a higher performance index than the other two functions, because it considers that water age performance is only null when it is higher than 48 h, and it assigns the maximum performance to all other water age values under 48 h. Conversely, the Coelho (1996) function is the most penalising to water age performance due to the low upper limit for water age (10 h). Surprisingly, the Nyirenda and Tanyimboh (2020) function, developed for storage tanks, gives PI values very similar to those obtained with the developed per-

formance curves for Case Study 1. This is likely due to a balance between not penalising nodes where water age is in the 21–48 h (winter) or 16–48 h (summer) range, which have null performance in the developed functions, and linearly penalising the nodes where water age is in the range of 8–17 h (winter) or 9–14 h (summer), which have maximum performance in the developed functions.

Overall, the results show that very different results can be obtained using the different performance functions. A system can be evaluated as performing very well or in an unacceptable way depending on the performance function used, which justifies the need for system-specific function development.

The developed performance functions and the obtained global performance index allowed us to quantify the performance of both case studies regarding water age. Such functions can also be incorporated into performance assessment systems for attending water quality in those analyses. Applying the penalty functions to all the consumption nodes allows for assessing water quality in the network, rather than evaluating the results of water quality analyses obtained for a small set of samples, which can hardly be representative of the whole system.

## 6. Conclusions and Further Research

In this paper, a novel methodology for developing water age performance functions is proposed and demonstrated in two water distribution systems in two seasons. The methodology is based on the establishment of a system-specific relationship between water age and chlorine concentrations and on a general chlorine performance function. This methodology, though based on chlorine content, does not require additional sampling, experimental decay tests, or water quality models other than those routinely carried out in the systems for water quality control purposes. It can be applied by making use of the chlorine data that the utilities gather for mandatory water quality control programmes and the simulated water age at the sampled nodes.

The developed performance functions were applied to two real case studies in two seasons. The obtained results were compared with those obtained by published performance functions, and major improvements were attained in comparison with the latter. This is because the water age limits used in the performance functions of the current study are tailored to the specifics of the analysed networks. These results highlight that network-specific water age performance functions can be a useful tool for more robustly and reliably globally assessing the performance of the systems regarding water age and for defining water age goals, which can be used in water age optimisation studies.

**Author Contributions:** Conceptualisation and methodology, D.C. and L.M.; writing—original draft preparation, R.A. and L.M.; writing—review and editing D.C. and L.M.; project management, D.C.; funding acquisition, D.C. All authors have read and agreed to the published version of the manuscript.

**Funding:** This work was supported by the Fundação para a Ciência e a Tecnologia (FCT) through the IMIST (grant number PTDC/ECI-EGC/32102/2017) and the WISDom project (grant number DSAIPA/DS/0089/2018).

**Institutional Review Board Statement:** Not applicable.

**Informed Consent Statement:** Not applicable.

**Data Availability Statement:** The data presented in this study are available upon request from the corresponding author.

**Acknowledgments:** The authors acknowledge the Fundação para a Ciência e Tecnologia (FCT) for funding the project PTDC/ECI-EGC/32102/2017, IMIST—Improving Mixing in Storage Tanks for safer water supply as well as the project DSAIPA/DS/0089/2018, WISDom. The authors are thankful to the water utilities that provided the data used in this research, namely Serviços Municipalizados de Castelo Branco and Infraquinta, Empresa de infraestruturas da Quinta do Lago, EM.

**Conflicts of Interest:** The authors declare no conflict of interest. The funders had no role in the design of the study; in the collection, analyses, or interpretation of data; in the writing of the manuscript; or in the decision to publish the results.

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
