# Peer review of "Enhanced Water Age Performance Assessment in Distribution Networks"

_water, doi:10.3390/w13182574_

Round 1

Reviewer 1 Report

The paper deals with a highly important topic: the quality of water in water distribution network from the point of view of water age -  a criteria that was less analysed in researches.

The paper is based on a critical analysis of up to date literature and presents results beyond the state of the art. The results are based on intensive data collection (5 years and the results are discussed also in comparison with the most significant results already published.

The paper could be enriched with some details about the data aquisition system (architecture, aquisition frequency, sensors - accuracy).

Author Response

The paper could be enriched with some details about the data aquisition system) (architecture, aquisition frequency, sensors - accuracy).

Reply | We would like to thank the reviewer for his comments, in particular the latter suggestion of including more details of the data collection system. Indeed, water quality data have been collected in periodical campaigns carried out in the water distribution systems in which water samples were manually taken from access points at different locations of the network. This is the procedure that is commonly carried out in drinking water systems in Portugal to comply with regulatory requirements established by the water authority (ERSAR). This clarification has been added to section 4.2.

Reviewer 2 Report

Overall I liked the Authors' idea of an enhanced water age performance assessment benefitting from chlorine measurements in the WDN. However, I have the following major comments, for each of which I expect to receive a reply and some work in the text:

1 - Introduction. Though the literature review is ok, I would suggest inserting citations to the works "Drinking water temperature around the globe: Understanding, policies, challenges and opportunities" and "Modulating Nodal Outflows to Guarantee Sufficient Disinfectant Residuals in Water Distribution Networks", which deal with the relationship between water quality and temperature and with the Opportunity to increase flow velocity at dead-ends to improve water quality, respectively.

2 - section 2.3. At the moment the Authors state that they aim to search for correlations between water age and chlorine measurements. Indeed, they investigate into the dependance age/chlorine residual. In fact, they use the coefficient of determination to evaluate the extent to which the former variable depend on the latter, based on regressive relationships. Had the Authors analyzed correlations, they would have used the pearson or spearman coefficient. Please, clarify this aspect throughout the paper.

3 - The Authors use modelling to derive water age, to be analyzed as a function chlorine measurements in the WDN. Why don't they use the flow (routing and water quality) modelling to derive numerical values of nodal chlorine residuals, to match observations. What is the benefit of building the bridge water age/chlorine residual. Indeed, in some countries, such as the Netherlands, no disinfectant is used in WDNs. Other countries use different kinds of disinfectants. In all these cases, the approach outlined by the Authors is bound to fail.

Minor comments

4 - there are problems of format and indents throughout the paper

5 - lines 17-19 may be misleading, in that they may be misinterpreted as something bad for the novel approach

6 - typos in lines 27-28

7 - lines 37-38 "increase in"

8 - lines 153-155. I don't understant these lines

Author Response

Overall I liked the Authors' idea of an enhanced water age performance assessment benefitting from chlorine measurements in the WDN. However, I have the following major comments, for each of which I expect to receive a reply and some work in the text:

1 - Introduction. Though the literature review is ok, I would suggest inserting citations to the works "Drinking water temperature around the globe: Understanding, policies, challenges and opportunities" and "Modulating Nodal Outflows to Guarantee Sufficient Disinfectant Residuals in Water Distribution Networks", which deal with the relationship between water quality and temperature and with the Opportunity to increase flow velocity at dead-ends to improve water quality, respectively.

Reply | These references have been added to the literature review in the introduction of the manuscript (references 12 and 13).

2 - section 2.3. At the moment the Authors state that they aim to search for correlations between water age and chlorine measurements. Indeed, they investigate into the dependance age/chlorine residual. In fact, they use the coefficient of determination to evaluate the extent to which the former variable depend on the latter, based on regressive relationships. Had the Authors analyzed correlations, they would have used the pearson or spearman coefficient. Please, clarify this aspect throughout the paper.

Reply | The Pearson coefficient is the square root of the determination coefficient, R2, thus, these two coefficients are related and using R2 indirectly reflects the value of Pearson coefficient. Additionally, the Pearson and Spearman coefficients are typically used to assess the correlation between parameters and not the quality of fitting of a regression law. In the current case, we wanted to obtain the mathematical law that described the water age variation with the Chlorine concentration, thus, R2 was used to measure the quality of this fitting.

3 - The Authors use modelling to derive water age, to be analyzed as a function chlorine measurements in the WDN. Why don't they use the flow (routing and water quality) modelling to derive numerical values of nodal chlorine residuals, to match observations. What is the benefit of building the bridge water age/chlorine residual. Indeed, in some countries, such as the Netherlands, no disinfectant is used in WDNs. Other countries use different kinds of disinfectants. In all these cases, the approach outlined by the Authors is bound to fail.

Reply | The main reason for the use of modelled water age, instead of modelled chlorine residuals, is to simplify the methodology in order to make it more easy for water utilities to use. While modelling water age only requires flow and pressure data, chlorine residuals modelling requires that same data plus the determination of a chlorine bulk decay coefficient (or more, depending on chosen chlorine decay kinetics) and the calibration of a wall decay coefficient. The bulk decay coefficient must be estimated based on laboratory decay tests (using the bottle tests), that are not routinely performed in water utilities laboratories or in other laboratories specialized in water analysis. Also, in many water utilities, there is no expertise on water quality modelling, and only the flow is modelled. 

Thus, the methodology developed makes the bridge between water age and chlorine residuals, using the already available data on chlorine content at the nodes, without requiring additional data or expertise that could hamper the methodology’s wider application.

Such an approach cannot be applied in the Netherlands or in any other country where chlorine is not added to the water (which are in less number that those that use chlorine as disinfectant), but similar approaches can be attempted, making use of other relevant water quality parameters that are monitored with high frequency. 

Minor comments

4 - there are problems of format and indents throughout the paper

Reply | These have been revised; the publisher will make the final revision

5 - lines 17-19 may be misleading, in that they may be misinterpreted as something bad for the novel approach

Reply | This sentence has been reformulated

6 - typos in lines 27-28

Reply | These have been corrected.

7 - lines 37-38 "increase in"

Reply | This has been corrected.

8 - lines 153-155. I don't understant these lines

Reply | The sentence was rewritten and a new one was added. We hope it is now clearer.

Reviewer 3 Report

Article "Enhanced water age performance assessment in distribution networks" presents an interesting method to build a correlation between model-derived water age and chlorine residual - physical parameter measured by water utilities to assess the microbiological stability of the water.

Authors present very briefly methodology for building the correlation and have shown the application of the method for two case studies in Portugal.

Based on the current use of water age as an indicator for water network optimisation studies it is refreshing to see the approach which is using online water quality monitoring data instead of regressing from time-domain only.

There are several shortcuts Authors made to apply the method, which requires clarification.

  1. Readers are lacking information on the quality of the model calibration. Especially handing and representing customers water demands throughout the seasons. 
  2. Due to variations in water consumption, is it required for the hydraulic model to be based on smart water metering data repositories instead of fixed multiplication approach from historical recordings? 
  3. To allow a wider scope of application Authors should explain more about free and total chlorine measurements. Many water utilities converted the disinfection method from chlorination to chloramination with UV. Measuring free (named also as residual) chlorine is not going to give correct information about the disinfection performance of water.
  4. Authors should address the effect of temperature on residual chlorine readings. Do you need to add a corrective factor to equalize results across seasons of the year?
  5. Both abstract and conclusions are lacking highlights (what is the added value for water utilities?) for the method based on results obtained in the case studies presented.

Author Response

There are several shortcuts Authors made to apply the method, which requires clarification.

1. Readers are lacking information on the quality of the model calibration. Especially handing and representing customers water demands throughout the seasons. 

Reply | A paragraph on model calibration was added.

2. Due to variations in water consumption, is it required for the hydraulic model to be based on smart water metering data repositories instead of fixed multiplication approach from historical recordings? 

Reply | Water consumption in the hydraulic model of case study 1 is described by a nodal consumption multiplied by a daily demand pattern, since there are no smart meters in this network. On the contrary, in case study 2, there are smart meters at the consumer level that allow the measurement of hourly consumption; in this case, the 24h-consumption values for the 2000 consumers have been introduced in the model; these values correspond to average values of data records to week days.

3. To allow a wider scope of application Authors should explain more about free and total chlorine measurements. Many water utilities converted the disinfection method from chlorination to chloramination with UV. Measuring free (named also as residual) chlorine is not going to give correct information about the disinfection performance of water.

Reply | The methodology developed is based on the existing data. Because in Portugal there is no chloramination and the nitrogen contents are very low, combined chlorine is not found in drinking water, only free chlorine. We agree that total chlorine would be the most suitable parameter to use as an indicator of disinfectant residual on chlorinated waters. A sentence was added in the manuscript.

4. Authors should address the effect of temperature on residual chlorine readings. Do you need to add a corrective factor to equalize results across seasons of the year?

Reply | Chlorine residual is measured by the DPD colorimetric method using very reliable kits for on-site determination. Though water temperature affects chorine decay rates, it does not affect chlorine readings when using the DPD colorimetric method and there is no need to add corrective factors (as for other water quality parameters). More detailed information on the DPD method can be found here: http://stpnq.com/wp-content/uploads/2014/08/Chlorine-Analysis-EN.pdf

5. Both abstract and conclusions are lacking highlights (what is the added value for water utilities?) for the method based on results obtained in the case studies presented.

Reply | Both the abstract and the conclusions have been revised to attend this comment.

Round 2

Reviewer 2 Report

All my comments have been implemented